## RESEARCH ARTICLE

# Coincident evolution and functional adaptation of the taxonomically restricted genes *ivph-3* and *gon-14* in *Caenorhabditis* nematodes

Nikita Jhaveri[1,*], Bhagwati Gupta[1] and Helen M. Chamberlin[2,‡]

## ABSTRACT

Poorly conserved or taxonomically restricted genes represent a sizable portion of most genomes. Many of these genes participate in essential processes and can contribute to evolutionary innovation in species. Here, we evaluate two of the *Caenorhabditis*-restricted class of LIN-15B-domain-encoding genes, *ivph-3* and *gon-14*, and compare their *in vivo* functions in two species, *C. elegans* and *C. briggsae*. We show that within the Elegans supergroup, *ivph-3* and *gon-14* exhibit sequence constraints distinct from other family members, including maintenance of a one-to-one orthology and a higher degree of sequence conservation. Coincidentally, mutants for either of the genes exhibit strong phenotypic defects that are similar within species (*C. elegans* or *C. briggsae*), but with notable differences when comparing between species. These findings highlight the genetic and genomic features associated with the evolution of a taxonomically restricted gene family.

KEY WORDS: Taxonomically restricted genes, Gene adaptation, *Caenorhabditis* nematodes

## INTRODUCTION

The principles of gene sequence conservation and orthology are fundamental to our understanding of biological systems and underlie the powerful tools we use to assign and infer function across genomes, proteins and phyla (Ashburner et al., 2000). While a high degree of gene sequence conservation across species suggests functional importance, it has long been clear that the obverse – that lack of sequence or phylogenetic conservation suggests functional unimportance – does not hold (Khalturin et al., 2009; Chen et al., 2013; Kondo et al., 2017; Baalsrud et al., 2018). Technical innovations and new data-mining methodologies provide one important approach to identifying gene families and prospective functions for 'orphan' genes and others that represent the genomic and proteomic 'dark matter', genes whose roles remain elusive due to limited conservation, unique evolutionary trajectories, or

[1]Department of Biology, McMaster University, Hamilton, ON L8S 4L8, Canada. [2]Department of Molecular Genetics, Ohio State University, Columbus, OH 43210, USA.
*Present Address: Department of Biology, Johns Hopkins University, Baltimore, MD, USA.

‡Author for correspondence (chamberlin.27@osu.edu)

N.J., 0009-0008-6434-2188; B.G., 0000-0001-8572-7054; H.M.C., 0000-0001-7203-2691

insufficient functional annotation (Toll-Riera et al., 2009; Durairaj et al., 2023; McKnight, 2024; Pereira et al., 2025). However, understanding the emergence, evolution, and function of poorly conserved genes remains an important unanswered question.

Here, we investigate the evolutionary origins and *in vivo* genetic characteristics of two *Caenorhabditis*-restricted genes, *ivph-3* and *gon-14*. *ivph-3* and *gon-14* are both members of a gene family that encode proteins with the RNAse H-fold-containing LIN-15B domain, a domain that may facilitate a protein's association with DNA and transcriptional repression (Rechtsteiner et al., 2019; Gal et al., 2021). LIN-15B-encoding genes are apparent only in a subset of *Caenorhabditis* species. Within the Elegans supergroup, *ivph-3* and *gon-14* are distinct from other family members in that they exhibit a higher degree of sequence conservation and retain a one-to-one orthology, suggestive of coincident adaptation of these two genes. We show that genetic disruption of *Cbr-ivph-3* results in widespread perturbation of gene transcripts, suggesting the gene extensively impacts both positive and negative gene regulation. Further, we show that disruption of either of these genes has strong and similar within-species phenotypic consequences in two species (*C. elegans* or *C. briggsae*), but that there are notable differences when comparing between species. Cross-species rescue experiments indicate that these changes include both gene-specific and species-specific differences. Altogether, this work reveals genetic and genomic hallmarks of new gene family emergence, and highlights changes that correlate as specific family members integrate into and support essential processes.

## RESULTS AND DISCUSSION

### RNA transcripts exhibit widespread dysregulation in *Cbr-ivph-3* mutants

Functions for the *ivph-3* (also known as *ivp-3*) gene were first understood in the nematode *C. briggsae*, as mutant alleles were recovered in genetic screens for animals with a Multivulva (Muv) phenotype (Sharanya et al., 2015; Chamberlin et al., 2020). While this phenotype is the most obvious in mutants, the animals exhibit several additional defects including alterations to mobility and sensitivity to oxidative stress (Fig. S1). To better understand the function of *Cbr-ivph-3*, we performed RNAseq experiments using RNA recovered from synchronized L3-stage larval animals (Tables S1-S5). Initial analysis revealed differential expression of 3852 genes, with a roughly equal distribution of upregulated (1898) and downregulated (1954) genes (Fig. 1A). We compared the *Cbr-ivph-3* data to previously reported datasets derived from two other *C. briggsae* mutants that exhibit the Muv phenotype and encode proteins that act to repress transcription of genes in *C. elegans* (*Cbr-spr-4(gu163)* encoding a C2H2 zinc finger protein and *Cbr-htz-1(gu167)* encoding H2A.z histone; Lakowski et al., 2003; Latorre et al., 2015; Chamberlin et al., 2020) to ask whether mutants for all three genes exhibit common

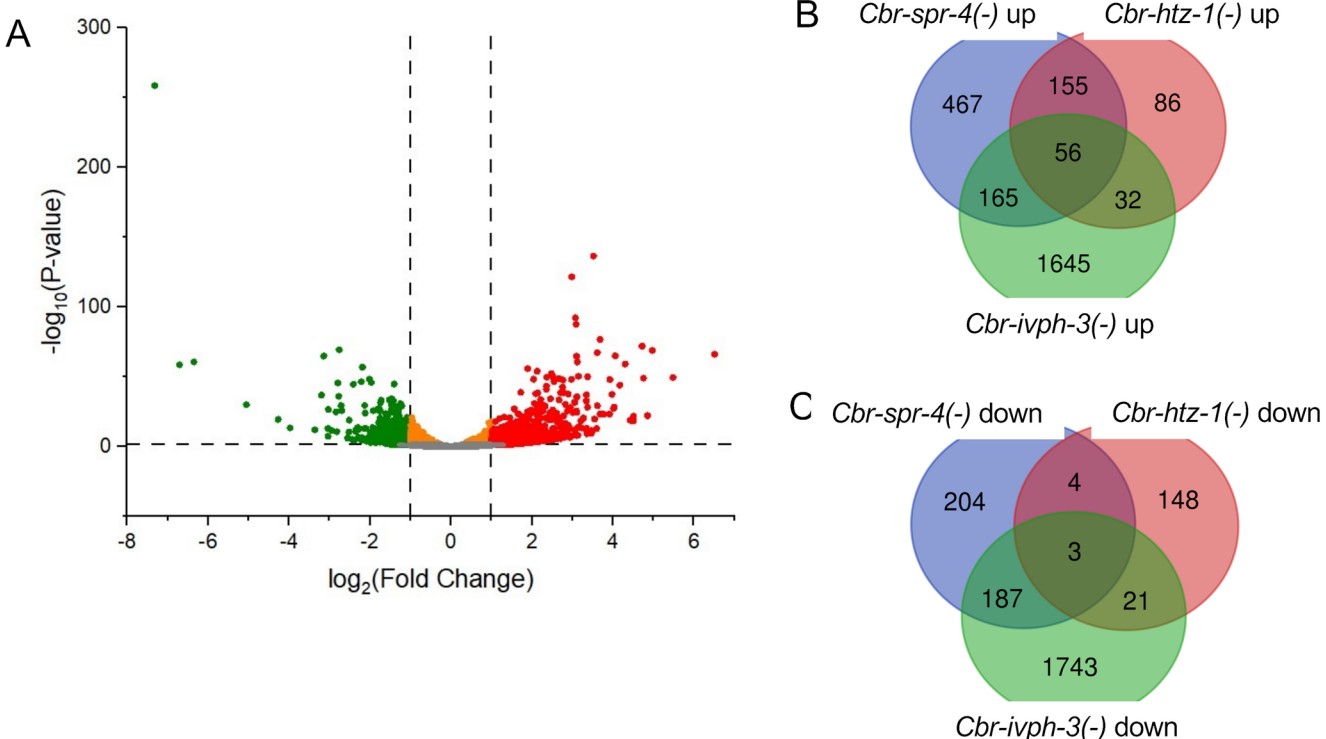

**Fig. 1. RNA transcripts exhibit widespread dysregulation in *Cbr-ivph-3* mutants.** (A) Volcano plot showing the differentially expressed (DE) genes in *Cbr-ivph-3(sy5216)* mutant transcriptome compared to AF16 wild type. 1898 genes exhibit increased abundance (log$_2$ fold change >+1) and 1954 genes exhibit decreased abundance (log$_2$ fold change <−1), or 49% of DE genes are increased compared to wild type. This is in contrast with the data from the Muv mutants *Cbr-spr-4(gu163)* and *Cbr-htz-1(gu167)* (Chamberlin et al., 2020) where the corresponding values are 843 and 398 (68% increased) for *Cbr-spr-4(gu163)* and 329 and 176 (65% increased) for *Cbr-htz-1(gu167)*, values that are statistically different from the *Cbr-ivph-3(sy5216)* data, but not each other (*P*<0.05, two-tailed proportional z-test). (B) Overlap of upregulated genes present in transcriptomic data sets of *Cbr-spr-4(gu163)*, *Cbr-htz-1(gu167)* and *Cbr-ivph-3(sy5216)*. Fifty-six genes exhibit increased abundance in all three mutants, including *Cbr-lin-3/EGF* that is confirmed previously for all three strains using RT-qPCR (Chamberlin et al., 2020). (C) Overlap of downregulated genes present in transcriptomic data sets of *Cbr-spr-4(gu163)*, *Cbr-htz-1(gu167)* and *Cbr-ivph-3(sy5216)*. Only three genes exhibit decreased abundance in all three mutants. Gene Ontology (GO) analysis of the DE genes common to all three strains identified positive regulation of mitotic nuclear division (GO:0045840) and positive regulation of nuclear division (GO:0051785). Data are in Tables S1-S5 (DE gene lists and overlaps) and S12-S15 (GO term analysis).

characteristics with respect to gene dysregulation. First, notably more genes exhibited differential expression (DE) in *Cbr-ivph-3* mutants compared to the other two (3852 versus 1241 and 505, respectively). In addition, upregulated genes represent a greater fraction of DE genes in *Cbr-spr-4* and *Cbr-htz-1* mutants, significantly different from *Cbr-ivph-3* (Fig. 1B,C). Previous experiments indicated that the Muv phenotype associated with *Cbr-ivph-3* mutants and others of this class results from increased and ectopic expression of the *Cbr-lin-3/EGF* gene (Chamberlin et al., 2020), and this gene is indeed represented among transcripts that exhibit increased abundance in the overlap of *Cbr-spr-4*, *Cbr-htz-1* and *Cbr-ivph-3* mutant strains (Table S4). Taken together, the results indicate that while *Cbr-ivph-3* shares some gene regulatory network characteristics with *Cbr-spr-4* and *Cbr-htz-1*, it is functionally distinct, impacting the transcripts of a broader number of genes with both positive and negative effects.

### IVPH-3 and other LIN-15B-domain-containing proteins are encoded by a gene class prevalent in a subset of *Caenorhabditis* species

*Cbr-ivph-3* has been identified to be part of a gene family that encodes proteins with the LIN-15B domain. This domain extends over 500 amino acids, and includes a sub-region with sequences similar to hAT transposases (Chesney et al., 2006; Chamberlin et al., 2020; Fig. S2). LIN-15B-domain-encoding genes are prevalent in the genomes of some *Caenorhabditis* species, with seven clear

representatives in *C. elegans* and six in *C. briggsae* (Fig. 2). However, there are no clear LIN-15B-domain containing proteins made in organisms with other highly curated genomes, such as human, mouse, zebrafish, or *Drosophila* (using Blastp; Altschul et al., 1990). While lack of Blastp hits does not rule out the possibility for highly divergent family members produced by a genome, these analyses support the classification of LIN-15B-domain encoding genes as lineage-restricted within *Caenorhabditis*, or poorly conserved across species.

To better understand the evolution of this gene family within nematodes, we evaluated sequence data from 16 *Caenorhabditis* species, distributed across the genus (Stevens et al., 2019, 2020; Sun et al., 2022). While multiple LIN-15B-domain-encoding genes are present in genomes derived from Elegans supergroup species and some related species, they are not detected in more basal *Caenorhabditis*, or in outgroup nematodes *Diploscapter pachys*, *Diploscapter coronatus*, or *Ascaris suum* (Fig. 2A,B). To ask whether structurally similar proteins may be encoded by other genomes but not apparent at the primary sequence level using Blastp, we used AlphaFold to evaluate predicted structural features (Jumper et al., 2021; Varadi et al., 2024). LIN-15B-domain-containing proteins all have many high confidence domains (pLDDT>90) including a canonical RNAse H fold (Fig. 2C; Fig. S2). Human proteins with similar predicted structures identified using Foldseek (AFDB50, van Kempen et al., 2024) include zinc finger BED-domain containing

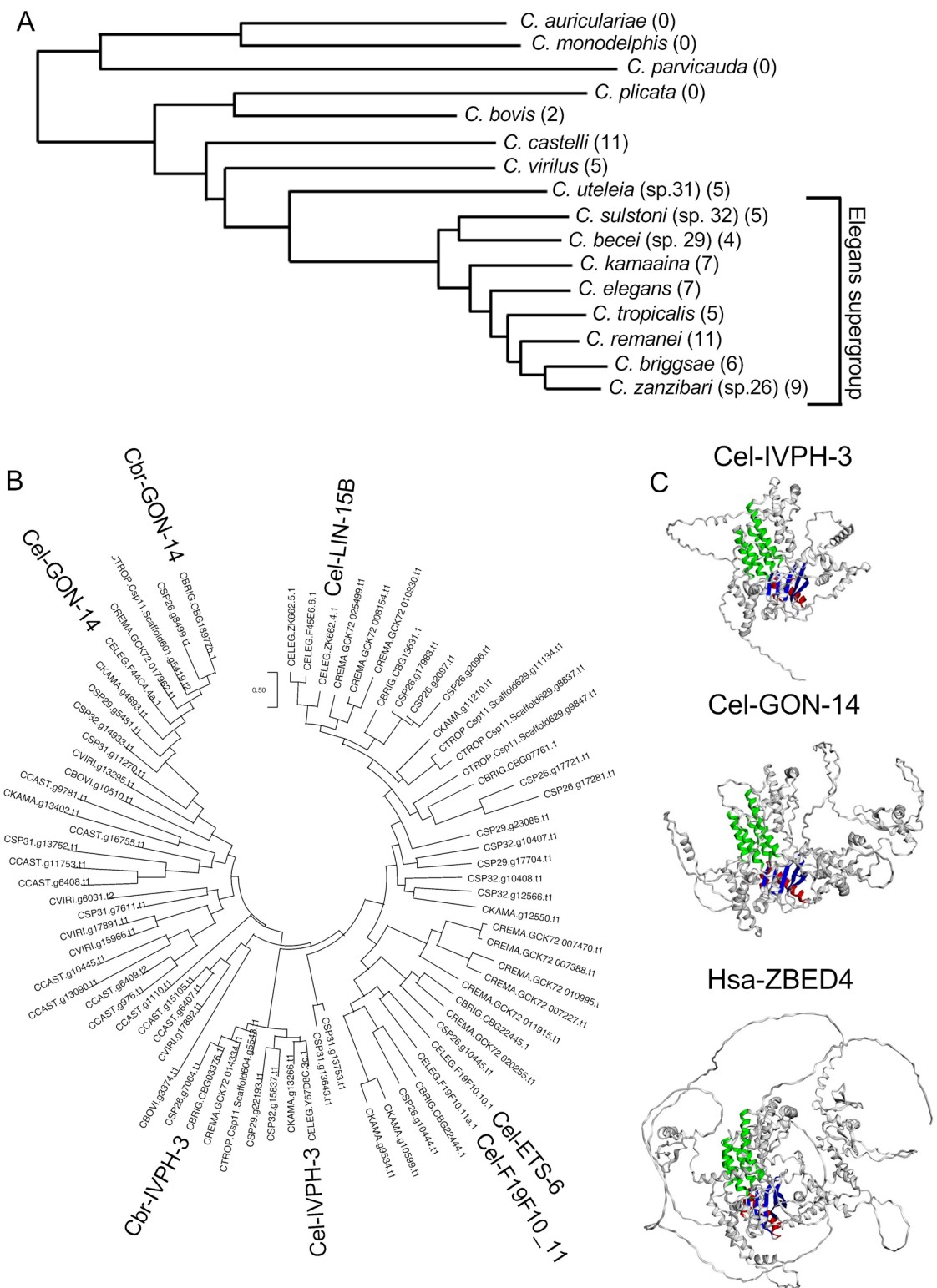

**Fig. 2. Multiple genes encoding LIN-15B domain proteins are present in the genomes of *Caenorhabditis* species of the Elegans-supergroup and related nematodes, but not apparent in more distantly related *Caenorhabditis* species.** (A) A phylogenetic tree for 16 *Caenorhabditis* species, adapted from Sun et al., 2022. The number of genes encoding proteins with a LIN-15B domain for each genome is listed in parentheses, based on Blastp hits at Caenorhabditis.org using default parameters. (B) Maximum likelihood tree of proteins with a LIN-15B domain from the species in panel A. *C. elegans* and *C. briggsae* IVPH-3 and GON-14 are highlighted, along with the three additional *C. elegans* protein sequences included in the alignment of Fig. S2. Scale is substitution per site. LIN-15B, F19F10.11 and ETS-6 also contain one or more THAP or THAP-like domains, but these are distinct and separate from the LIN-15B domain (Clouaire et al., 2005). (C) Predicted structures for Cel-IVPH-3 (AF-Q86DD2-F1-v4), Cel-GON-14 (AF-Q4PIU4-F1-v4), and Hsa-ZBED4 (AF-O75132-F1-v4) from AlphaFold (Jumper et al., 2021), annotated with the color coding of Fig. S2 and orientation adjusted using EzMol (Reynolds et al., 2018).

proteins (ZBED) (Fig. 2C; Table S17). We conclude that LIN-15B-domain containing proteins share structural features with proteins from other species, but they lack clear orthologs outside of a subset of *Caenorhabditis* species. While the origin of these *Caenorhabditis* genes is not clear, it is notable that genes encoding structurally similar proteins, including *ZBED* genes in vertebrates and *SLEEPER* genes in angiosperms, originated as transposons that subsequently were 'domesticated', with their products adapted to critical roles in a variety of important host cell processes (Sinzelle et al., 2009; Arensburger et al., 2011; Knip et al., 2012; Hayward et al., 2013).

### *ivph-3* and *gon-14* exhibit distinct sequence constraints in Elegans supergroup species compared to other LIN-15B-domain-encoding genes

We constructed a phylogenetic tree of the LIN-15B-domain-containing proteins to better understand the emergence and relationships among the encoding genes in *Caenorhabditis* (Fig. 2B). Our analysis revealed that the genomes exhibit a high degree of tolerance to duplication, loss, and sequence change for most genes. For example, the *C. elegans* genome contains seven genes in this family, five of which are *lin-15B* and two partial duplicates of the *lin-15B* gene, as well as the duplicate genes *Cel-ets-6* and *Cel-F19F10.11* (Fig. 2B). In contrast, among genomes of the Elegans supergroup species, genes encoding IVPH-3- or GON-14-related proteins retain an apparent one-to-one orthology and a relatively high level of sequence conservation compared to other family members (Fig. 2B). We conclude that *ivph-3* and *gon-14* exhibit gene copy number and sequence constraints distinct from the other family members. We hypothesize that these features, when observed in a novel or poorly conserved gene family, correlate with an adaptive transition of gene function. For example, such selection modifications may occur when a gene's product embeds in a biological process where enzymatic functions or interactive partners place greater constraints on stoichiometry and sequence characteristics (Worth et al., 2009). The coincident behavior for two genes in the family (*ivph-3* and *gon-14*) is suggestive that they adapted to a common process.

### *ivph-3* and *gon-14* mutants exhibit similar within-species phenotypes, and coincident evolution of phenotype

To better understand the relationship between *ivph-3* and *gon-14* and how these genes function *in vivo*, we evaluated mutant phenotypes in *C. elegans* compared to *C. briggsae* (Fig. 3). *Cel-gon-14* mutants were isolated in genetic screens for hermaphrodites with defects in specification of the somatic gonad distal tip cells (DTCs) which are essential for proper development of the gonad and the support of germ cell proliferation (Siegfried et al., 2004; Chesney et al., 2006). Consistent with this function, homozygous *Cel-gon-14* mutant hermaphrodites are sterile, and frequently lack one or both gonad arms (Fig. 3B,D-E; Siegfried et al., 2004). Many *ivph-3(gk3691)* mutants are sterile, but most derived from heterozygous mothers retain both gonad arms (Fig. 3C-E). Any offspring from fertile *ivph-3(gk3691)* homozygous mutants are sterile (Maternal Effect Sterile, Mes, Fig. 3E), and all exhibit defects in gonad arm outgrowth (Fig. 3D). We conclude that fertility and gonad arm development defects are shared phenotypes associated with *Cel-ivph-3* and *Cel-gon-14* mutants. In contrast to the *C. elegans* mutants, *Cbr-ivph-3* and *Cbr-gon-14* mutants are generally fertile (Fig. 3E, Chamberlin et al., 2020). For both *C. elegans* and *C. briggsae, ivph-3; gon-14* double mutants do not exhibit an enhanced phenotype compared to single mutants, a result consistent with the idea that in both species, the two genes

participate in a common process. Homozygous *Cbr-ivph-3* or *Cbr-gon-14* mutants exhibit a strong Muv phenotype (Fig. 3F; Sharanya et al., 2015; Chamberlin et al., 2020). By contrast, *Cel-ivph-3* and *Cel-gon-14* mutants are generally nonMuv with a normal pattern of vulva development, and this phenotype is not enhanced in double mutants (Fig. 3F-G; Chesney et al., 2006; Chamberlin et al., 2020).

### Species differences in function for *ivph-3* and *gon-14* reflect both context-specific and gene-specific changes

To identify the source of the phenotypic differences between the two species, we evaluated the capacity of each gene to rescue genetic mutants. We introduced wild-type genomic DNA PCR fragments as transgenes, and asked whether they can rescue the mutant phenotype (Fig. 4). *Cbr-gon-14(+)* DNA rescues *gon-14* mutants of either species, whereas *Cel-gon-14(+)* DNA only rescues *Cel-gon-14* mutants (Fig. 4B,D). By contrast, *Cel-ivph-3(+)* DNA rescues *ivph-3* mutants of either species, whereas *Cbr-ivph-3(+)* DNA only rescues *Cbr-ivph-3* mutants (Fig. 4B,D). Since *Cel-ivph-3* and *Cbr-ivph-3* differ in that *Cel-ivph-3* can produce transcripts with upstream exons not present in *Cbr-ivph-3* (Fig. S3; wormbase.org Davis et al., 2022), we hypothesized that this difference might reflect *Cel-ivph-3*-specific functions for the longer transcript. However, either a fragment including all *Cel-ivph-3* exons (*Cel-ivph-3(+)L*) or one that lacks the two most 5′ exons (*Cel-ivph-3(+)S*) is sufficient to rescue the Ste phenotype associated with *Cel-ivph-3(gk3691)* (Fig. 4B). Altogether, we conclude that the *Cel-gon-14(+)* or *Cbr-ivph-3(+)* sequences sufficient for function within species are not sufficient to rescue across different species, whereas those of *Cbr-gon-14(+)* and *Cel-ivph-3(+)* are. These results identify structural and functional differences between *gon-14* and *ivph-3* in the two species. Additionally, cellular and genetic context plays a role in the ultimate phenotypic consequences, since loss of *Cbr-gon-14* has a minor impact on fertility and loss of *Cel-ivph-3* has a minor impact on vulval development, but the wild-type gene can fully restore these functions when introduced into cross-species mutants with the defect.

One explanation for the cross-species gene rescue results could be that the genes capable of rescuing in both species are expressed and function in a broader set of cells compared to the genes with more restricted rescue ability. Single cell RNAseq data indicate that the expression pattern for each gene is relatively well conserved, and both genes have relatively broad expression in both species (Large et al., 2025; Toker et al., 2025). Indeed, a cell-type expression specificity measure (Tau) indicates that in both cases the gene with ability to rescue in both species exhibits a greater cell-restricted pattern of expression than its ortholog (Large et al., 2025; Table S19). We evaluated the rescue ability of chimeric genes composed of sequences upstream of the start codon from one species and downstream sequences from the other, and found that while either chimeric transgene can rescue the mutant phenotype for strains rescued by the genomic DNA of both species, *Cbr-gon-14* mutants were poorly rescued by both, and only the *Cbr_Cel-ivph-3(+)* transgene (upstream sequences from *Cbr-ivph-3* and downstream from *Cel-ivph-3*) restored fertility to the *Cel-ivph-3* mutants (Fig. 4C,E). Altogether, these results indicate there may be multiple distinctions between the species for each gene, although future experiments will be necessary to identify their specific differences.

Taxonomically restricted genes are prevalent in all genomes and have a variety of evolutionary origins (Pereira et al., 2025). They are of evolutionary interest as a class due to their capacity to be recruited to essential biological processes *de novo*, and to contribute to novel

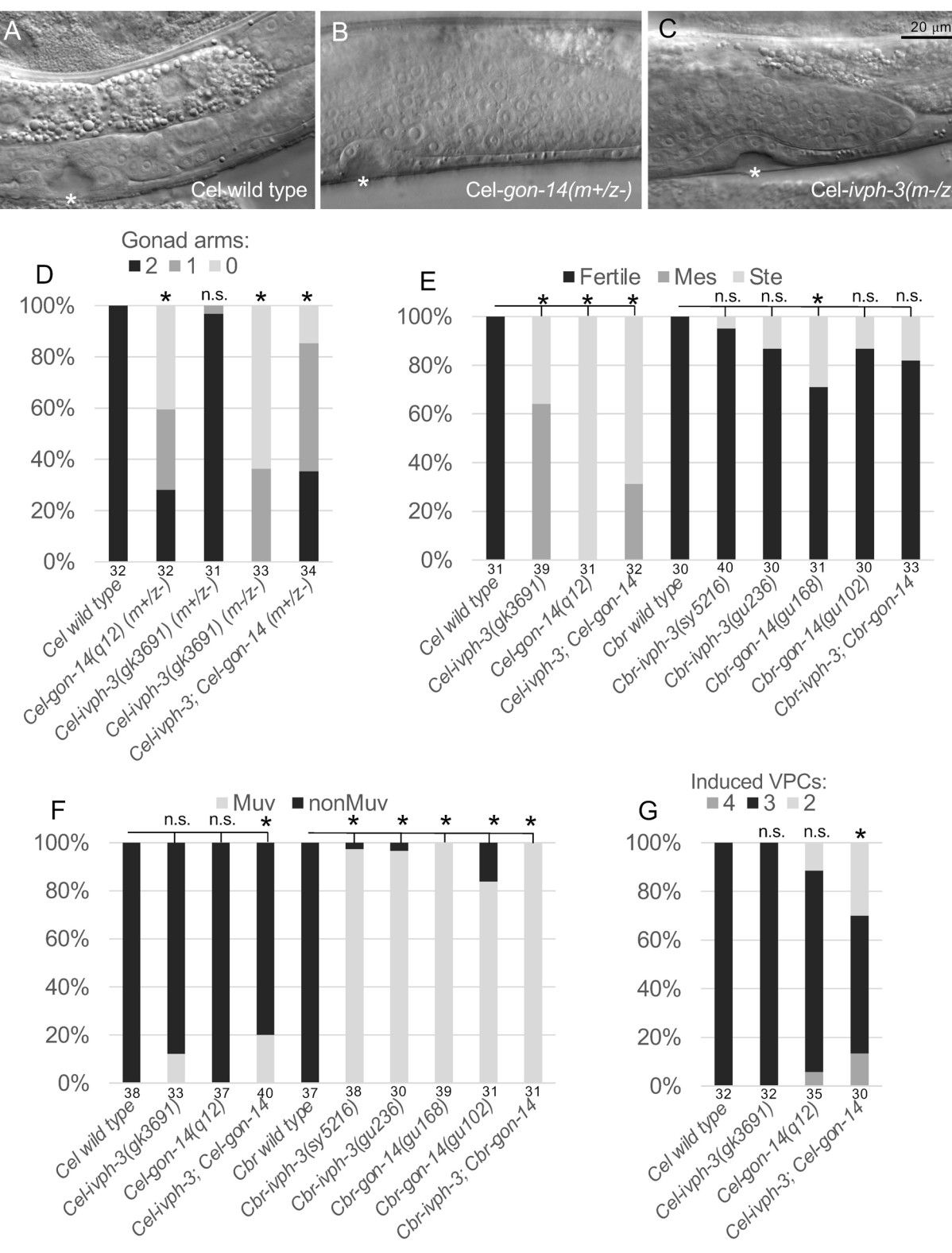

**Fig. 3.** See next page for legend.

biological adaptations (Khalturin et al., 2009). Here, we evaluate the characteristics for one taxonomically restricted gene class, those encoding LIN-15B-domain proteins in *Caenorhabditis*. The LIN-15B domain includes an RNAse H fold, an evolutionary ancient protein structure that is prevalent in transposases and other nucleases (Ma et al., 2008; Nowotny, 2009; Majorek et al., 2014).

While we do not rule out the possibility that IVPH-3 and GON-14 function as nucleases, other RNAse H fold-containing proteins such as those in the ZBED family associate with specific DNA elements and act as transcriptional regulators (Markljung et al., 2009; Hayward et al., 2013; Somerville et al., 2020). In *C. elegans*, LIN-15B itself associates with specific sites in the genome, and impacts

**Fig. 3. *Cel-gon-14* and *Cel-ivph-3* mutants exhibit similar defects in gonad development and fertility, and are phenotypically distinct from *Cbr-gon-14* and *Cbr-ivph-3* mutants.** (A-D) Gonad arm outgrowth is defective in *Cel-gon-14* and *Cel-ivph-3* mutants. (A) In wild-type hermaphrodites, two gonad arms develop from the gonad primordium in the animal midbody. In L4 larvae, the arms have developed toward the animal ends, completed a U-turn, and migrated back toward the midbody along the dorsal side of the body cavity. Asterisk indicates the position of the developing vulva opening, and a single gonad arm (posterior) is shown. (B) *Cel-gon-14* mutants derived from heterozygous mothers exhibit defects in distal tip cell (DTC) differentiation and subsequent outgrowth of the gonad arms (Siegfried et al., 2004), and the gonad cells fail to migrate and frequently form a single mass of cells in the center of the animal. (C) While the gonads of *Cel-ivph-3* mutants derived from heterozygous mothers are generally normal in morphology (not shown), *Cel-ivph-3* mutant offspring produced by mutant mothers (m-z-) likewise exhibit gonad arm defects. (D) Quantification of the gonad arm defect. Introduction of *Cel-ivph-3(gk3691)* into the *Cel-gon-14(q12)* background does not enhance the phenotype. Asterisk indicates statistically different from Cel wild type (P<0.05, Fisher-Freeman-Halton test with Bonferroni correction). n.s., not significantly different. *Cel-ivph-3; Cel-gon-14* is also not significantly different from *Cel-gon-14(q12)* (using the same test). (E) Homozygous mutant *Cel-gon-14* or *Cel-ivph-3* animals fail to produce offspring (Ste) or produce some offspring that all fail to produce offspring (Mes). Double mutant animals exhibit a phenotype similar to the single mutants. Homozygous mutant *Cbr-gon-14* or *Cbr-ivph-3* animals are generally fertile, as are double mutants (Chamberlin et al., 2020). Asterisk indicates statistically different from the wild type (P<0.05, Fisher-Freeman-Halton test with Bonferroni correction). n.s., not significantly different. (F) Homozygous mutant *Cbr-gon-14* or *Cbr-ivph-3* mutants exhibit a strong Multivulva (Muv) phenotype, whereas *Cel-gon-14* and *Cel-ivph-3* mutants are generally nonMuv. Asterisk indicates statistically different from the wild type (P<0.05, Fisher's exact test with Bonferroni correction). n.s., not significantly different. (G) Animals that do exhibit ventral protrusions (scored as Muv) typically exhibit defects in vulva morphogenesis rather than inappropriate vulva cell proliferation, as only the wild-type of three vulva precursor cells (VPCs) typically divide to produce vulva cells in *Cel-gon-14* and *Cel-ivph-3* mutants. *Cel-gon-14* single and *Cel-ivph-3; Cel-gon-14* double mutants exhibit increased variation in the number of induced cells, with some animals producing two or four induced cells rather than three as in the wild type. Asterisk indicates statistically different from the wild type (P<0.05, Fisher-Freeman-Halton test with Bonferroni correction). n.s., not significantly different. Although the mutant strains have an increased frequency of non-wild-type cell patterns, the average number of induced cells (induction index) for single and double mutants is not different from wild type (t-test with Bonferroni correction). Average number of induced cells for these data: Cel wild type (3, s.d.=0), *Cel-ivph-3(gk3691)* (3, s.d.=0), *Cel-gon-14(q12)* (2.9, s.d.=0.416), *Cel-ivph-3; Cel-gon-14* (2.8, s.d.=0.648). Full genotypes for double mutants: *Cel-ivph-3(gk3691); Cel-gon-14(q12). Cbr-ivph-3(sy5216); Cbr-gon-14(gu102)*. Sample sizes are indicated at the base of each bar.

the transcription of target genes (Rechtsteiner et al., 2019; Gal et al., 2021), and alleles of *Cel-gon-14* have been identified in genetic screens for altered transcriptional response to mitochondrial dysfunction (Xu et al., 2024). However, the mechanisms for how GON-14 and IVPH-3 function remain important work for the future.

Our results indicate that *Caenorhabditis* LIN-15B domain-encoding genes exhibit distinct selection characteristics that correlate with *in vivo* function. Two genes, *ivph-3* and *gon-14*, each play an important role in development in two species, *C. elegans* and *C. briggsae*. Where tested in *C. elegans*, genetic disruption of others in the family do not exhibit overt phenotypes on their own under standard growth conditions, although several exhibit synthetic or redundant functions with other genes (Clark et al., 1994; Huang et al., 1994; Byrne et al., 2007; Sawyer et al., 2011), and *Cel-lin-15B* participates in a variety of gene regulatory processes (Wang et al., 2005; Petrella et al., 2011; Gal et al., 2021).

Overall, our analysis provides a deeper understanding of how novel taxonomically restricted genes transition from redundant or auxiliary roles to essential functions, shedding light on their evolutionary significance and their adaptation to species-specific roles.

## MATERIALS AND METHODS
### Worm maintenance and genetics

*Caenorhabditis* strains were grown on NGM (or NG-Agar) plates seeded with *Escherichia coli* strain OP50 as a food source (Stiernagle, 2006). All experiments were carried out at 20°C unless otherwise indicated. The wild-type *C. elegans* used was N2 Bristol. The wild-type *C. briggsae* used was AF16. Specific strains and genotypes are listed in Table S6. *Cel-gon-14(q12)* is a G to A substitution that results in a premature termination codon at amino acid 598 (exon 9) (Chesney et al., 2006). *Cel-ivph-3(gk3691) is* an allele that deletes exon 4 and the first nine bp of exon 5 of the longest transcript (Fig. 4; exon 1 and 2 of Y67D8C.3a.1; Chamberlin et al., 2020). *Cbr-gon-14(gu102), Cbr-gon-14(gu168), Cbr-ivph-3(sy5216)* and *Cbr-ivph-3(gu236)* each introduce a premature stop codon into their respective genes (Chamberlin et al., 2020). *Cbr-gon-14(gu102)* is a G to A substitution that alters the 5′ splice recognition site of intron 8 (of CBG18977b.1), resulting in failure to remove the intron and introduction of an in-frame stop codon. *Cbr-gon-14(gu168)* is a C to T substitution that introduces a premature stop codon into exon 7. *Cbr-ivph-3(sy5216)* is a G to A substitution that introduces a premature stop codon into exon 7 (of CBG03376.1). *Cbr-ivph-3(gu236)* is a C to T substitution that introduces a premature stop codon into exon 4.

### RNA preparation, sequencing, and analysis

*C. briggsae* animals (AF16 wild type or PS9217 *Cbr-ivph-3(sy5216)*) were developmentally synchronized using a standard bleaching method (Porta-de-la-Riva et al., 2012). Synchronized L1s were plated onto OP50 seeded plates and incubated at 20°C. Larval animals were staged based on VPC division using Nomarski optics (Yochem, 2006; Chamberlin et al., 2020), and harvested for RNA following the Trizol (Invitrogen) protocol. Three independent biological replicates were prepared for each genotype and sequenced using the Illumina NovaSeq 6000 system by Genome Quebec to generate paired-end reads. The average number of reads obtained per sample was 2.9 million (ranging from 2.5 million to 3.4 million), with an average read length of 200 bp. RNA sequencing reads that were generated by Genome Quebec were processed using the online program, Galaxy (Galaxy Community, 2024, https://usegalaxy.org/). The algorithm for each program is outlined in Table S11. Gene Ontology (GO) enrichment analysis was performed using the PANTHER overrepresentation test (version 20240807; Mi et al., 2021) (Tables S12-S15).

### Evaluation of IVPH-3-related protein sequence

The amino acid sequence of Cel-IVPH-3 was obtained from wormbase.org (WS294) (Davis et al., 2022), and used in the protein blast function on Caenorhabditis.org against datasets for the 16 *Caenorhabditis* species listed in Fig. 2A. The resulting sequences were filtered to retain a single (the longest) isoform for each predicted protein hit (Table S9). The tree of Fig. 2B was produced using these sequences, aligned using Muscle and evaluated using the Jones Taylor-Thornton Maximum Likelihood method within MEGA11 (Tamura et al., 2021).

Four *C. elegans* proteins have both a full-length LIN-15B-domain and predicted structures represented in the AlphaFold database (Cel-IVPH-3, Cel-GON-14, Cel-LIN-15B, Cel-ETS-6). These all have an average predicted local distance difference test (pLDDT) value between 60-75, but with many high confidence domains (pLDDT>90). All four proteins are annotated for an RNAse H fold (TED classification 3.30.420, Nucleotidyltransferase; domain 5; Lau et al., 2024) followed by one or more unclassified domains that correspond to other elements of the LIN-15B domain (Fig. 2C; Fig. S2). Structurally related proteins for Cel-IVPH-3 (AF-Q86DD2-F1-v4) and Cel-GON-14 (AF-Q4PIU4-F1-v4) were identified using Foldseek and both the PDB and AFDB50 databases (van Kempen et al., 2024) as implemented on the Alphafold website

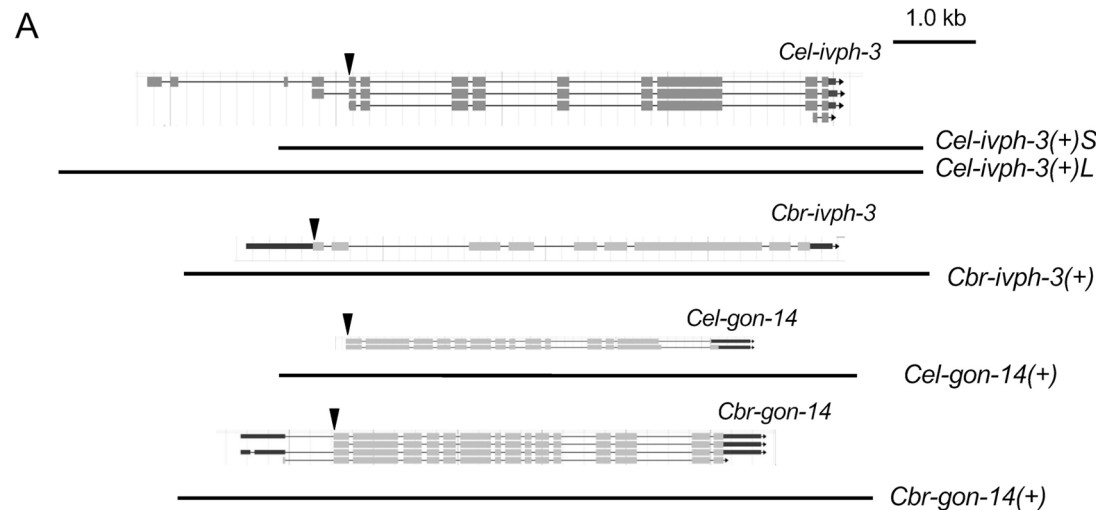

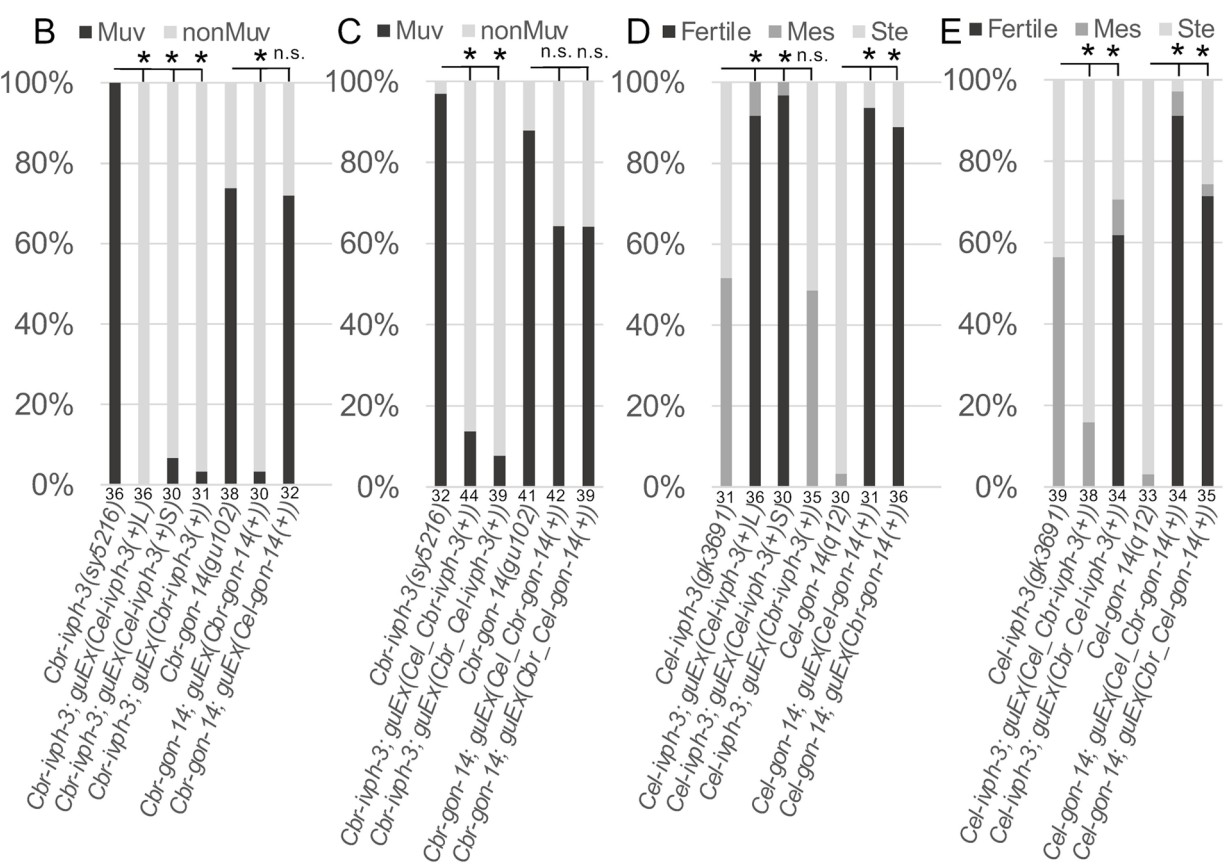

**Fig. 4. *ivph-3* and *gon-14* exhibit distinct capacities to compensate across species.** (A) Gene models for *Cel-ivph-3*, *Cbr-ivph-3*, *Cel-gon-14* and *Cbr-gon-14* (from wormbase.org, Davis et al., 2022), annotated to indicate the extent of genomic sequence included in the PCR products used for transgenic rescue in panels B-E. A triangle indicates the swap point for the chimeric genes evaluated in C and E. (B,C) Transgenic rescue experiments for *Cbr-ivph-3* and *Cbr-gon-14* mutants. (B) Either long (L) or short (S) *Cel-ivph-3(+)* DNA rescues the Muv phenotype associated with *Cbr-ivph-3* mutants, whereas *Cel-gon-14(+)* DNA does not rescue the phenotype in *Cbr-gon-14* mutants. (C) Either *Cbr_Cel-ivph-3(+)* DNA (*Cbr* sequences upstream and *Cel* sequences downstream of the gene start codon) or *Cel_Cbr-ivph-3(+)* DNA can rescue the Muv phenotype associated *Cbr-ivph-3* mutants. Neither *Cbr_Cel-gon-14(+)* DNA nor *Cel_Cbr-gon-14(+)* DNA restores gene activity as does the intact *Cbr-gon-14(+)* DNA in *Cbr-gon-14* mutants. (D,E) Transgenic rescue experiments for *Cel-ivph-3* and *Cel-gon-14* mutants. (D) Either *Cel-gon-14(+)* or *Cbr-gon-14(+)* DNA rescues the Ste phenotype associated with *Cel-gon-14* mutants. Either long (L) or short (S) *Cel-ivph-3(+)* DNA can rescue *Cel-ivph-3* mutants but *Cbr-ivph-3(+)* DNA does not. (E) *Cbr_Cel-ivph-3(+)* DNA can restore fertility to *Cel-ivph-3* mutants, whereas *Cel_Cbr-ivph-3(+)* DNA does not. *Cel_Cbr-ivph-3(+)* transgene-bearing animals are statistically different from control as more animals are Ste rather than Mes. Either the *Cbr_Cel-gon-14(+)* DNA or the *Cel_Cbr-gon-14(+)* DNA can restore fertility to *Cel-gon-14* mutants. Asterisk indicates statistically different from the homozygous mutant without a transgene (*P*<0.05, Fisher's exact test or Fisher-Freeman-Halton test, with Bonferroni correction). n.s., not significantly different. Sample sizes are indicated at the base of each bar.

(AlphaFold.ebi.ac.uk; Jumper et al., 2021). This identified structurally related proteins for Cel-IVPH-3 [22 in PDB and the maximum (1000) in AFDB50] and Cel-GON-14 (23 in PDB and 64 in AFDB50). Top hits for each outside of *Caenorhabditis* were proteins annotated as including a BED domain (sorted by lowest E-value). The full list of hits for each protein was interrogated with the same species or genera queried for Blast search [human (*Homo sapiens*), mouse (*Mus musculus*), zebrafish (*Danio*), fruit fly (*Drosophila*), nematodes (*Caenorhabditis*, *Ascaris* and *Diploscapter*)]. Proteins from none of these species are represented in the Cel-GON-14 sets. Within the Cel-IVPH-3-related sets, a single hit (human DNA mismatch repair protein Msh3) was identified in PDB (Table S16). Hits in the AFDB50 database include one in human (Zinc finger BED domain-containing protein 4), two in zebrafish, and seven in *Drosophila* species (two in *D. melanogaster*, and one each in five other species), all identified either as BED domain-containing or uncharacterized proteins (Table S17). No hits in either database were identified for the two outgroup nematode genera, although the AFDB50 database included over 27,000 structural model predictions from *Ascaris* (primarily *Ascaris lumbicoides*) and over 32,000 from *Diploscapter* (primarily *Diploscapter pachys*). Within *Caenorhabditis*, only hits from Elegans supergroup species were identified (Table S18), although very few predicted protein models are included for species outside of the Elegans supergroup in AFDB50. Predicted structures for Cel-IVPH-3 (AF-Q86DD2-F1-v4), Cel-GON-14 (AF-Q4PIU4-F1-v4), and Hsa-ZBED4 (AF-O75132-F1-v4) were obtained from AlphaFold.ebi.ac.uk (Jumper et al., 2021; Varadi et al., 2024). These structures were annotated with color to highlight structural features and adjusted for orientation using EzMol (www.sbg.bio.ic.ac.uk/ezmol/) (Reynolds et al., 2018).

### Molecular biology and transgenics
PCR was used to amplify products from wild type (*C. briggsae* AF16 or *C. elegans* N2) genomic DNA (primers in Table S8). These PCR products were incorporated into injection mixes (25 ng/µl PCR product, 15 ng/µl pDP#MM016 (*Cel-unc-119(+)*), 10-20 ng/µl pCFJ90 (*myo-2::mCherry*), and 50 ng/µl 1 kb ladder (NEB) as a carrier), and injected into the gonad of adult hermaphrodites following standard *C. elegans* protocols (Mello et al., 1991). Specific transgene details are in Table S7.

### Phenotypic assays
#### Fertility
L4 hermaphrodites ($P_0$) were selected individually onto plates and observed for offspring after 2 days and after 7 days. Animals with no apparent eggs or offspring on either day were scored as Sterile (Ste). Animals with apparent eggs and offspring on both days, with clear production of offspring from the F1 progeny, were scored as Fertile. Animals with apparent eggs or offspring on day 2, but no apparent progeny from the F1 animals were scored as Maternal Effect Sterile (Mes). All experiments in a given graph were conducted in parallel, in 2-4 batches.

#### Vulva development
Production of the Multivulva (Muv) phenotype was measured in young adult animals by selecting L4 hermaphrodites onto plates and observing their morphology on the next day (Lyu and Chamberlin, 2024). To measure the extent of vulva cell development (induction; Fig. 3G), early L4 larval animals were selected and evaluated for the number of vulval cells and inferred number of VPCs that had divided to produce them (Sternberg and Horvitz, 1986). All experiments in a given graph were conducted in parallel, in 2-4 batches.

#### Gonad morphology
Microscopic slides were prepared as described above, and gonad arm morphology was scored in L4 larvae. In wild-type *C. elegans* (and *C. briggsae*) hermaphrodites, the gonad develops with two arms that extend from the mid-body along the ventral of the animal, then migrate to the dorsal and back toward the mid-body. Gonad arms lacking a distal tip cell (DTC) fail to migrate. L4 animals were evaluated for the presence of gonad arms that had completed an extent of ventral migration, as well as migration to the dorsal. All experiments in a given graph were conducted in parallel, in 2-4 batches.

### Animal movement
Young adult animals (day-1 adult) were obtained by separating well-fed L4 animals to a fresh plate and performing the analyses on the next day. Pharyngeal pumping was measured in day-1 adult hermaphrodites under a dissecting microscope, counting the number of times pharyngeal contraction was observed in 30 s. Body bends were measured by counting the number of sinusoidal waves the worm made in 1 min (one wave corresponds to one body bend). The reading was not counted if the worm stopped moving before 1 min time elapsed.

### Oxidative stress
Methyl viologen (Paraquat, PQ) was used to assay oxidative stress. Young adult (day-1) worms were washed off a synchronized plate and added into 300 µl of 200 mM PQ in M9 in 24-well plates. 50–100 worms were added in each well, and two wells were used for each strain. Animals were evaluated once per hour over a 4 h period for their response to touch – if animals did not respond and had a rod-like appearance, they were counted as dead.

### Sequence alignment
The LIN-15B domains of *C. elegans* LIN-15B-related proteins were recovered and cropped from the *C. elegans* sequences in Table S9 (based on the blast alignment), yielding the sequences listed in Table S10. These sequences were aligned using the EMBL-EBI Clustal Omega server (https://www.ebi.ac.uk/jdispatcher/msa/clustalo) (Sievers et al., 2011), and shaded using BoxShade (https://junli.netlify.app/apps/boxshade/).

### *Cbr-ivph-3* cDNA
To define the transcripts made from the *Cbr-ivph-3* locus and to validate the predicted gene model, we harvested RNA from day-1 wild-type animals (Trizol), synthesized first strand cDNA using LunaScript RT Supermix kit (NEB #E3010) that contained random hexamers and oligo-dT primers and used PCR to amplify possible products with a *Cbr-ivph-3*-specific reverse primer in predicted exon 7, and primers to recognize splice leaders (SL1 and SL2). Two products were obtained using the SL1-specific primer. These products were separated on an agarose gene, excised, and subject to Sanger sequencing, which identified two SL1-spliced isoforms which include the longer isoform represented in the gene model, as well as an isoform that begins with exon 3 of the longer isoform. Primers are listed in Table S8.

### Acknowledgements
We thank A.T. Dawes for comments on the manuscript, and H. Lyu for technical assistance. Some strains were supplied by the *Caenorhabditis* Genetics Center, which is funded by the NIH Office of Research Infrastructure Programs (P40 OD010440). Sanger sequencing was performed by the OSUCCC Genomics Core Shared Resource, which is subsidized by an NIH Cancer Center Support Grant (P30CA016058).

### Competing interests
The authors declare no competing or financial interests.

### Author contributions
Conceptualization: N.J., B.G., H.C.; Data curation: N.J., H.C.; Funding acquisition: B.G., H.C.; Investigation: N.J., H.C.; Methodology: N.J., H.C.; Supervision: B.G., H.C.; Writing – original draft: H.C.; Writing – review & editing: N.J., B.G., H.C.

### Funding
This work was supported by a National Institutes of Health award to H.M.C. (R21OD030067), and Natural Sciences and Engineering Research Council of Canada Discover grant to B.P.G. Additional support was from the Mitcas Globalink Research Award, the Global Experience Award (McMaster University, Spring 2022), a Spring/Summer GSA Travel Assistance Award (Spring 2022), and McMaster SGS Grant in Aid of Travel Research and Field Study (Spring 2022), all awarded to N.J. Open Access funding provided by Ohio State University. Deposited in PMC for immediate release.

### Data and resource availability
Strains and plasmids are available upon request. Raw input data (genomes and gene annotations) can be retrieved from WormBase (under WS279 release) and NCBI (under PRJNA10731 and GSE288297 for RNA sequencing data sets of *Cbr-ivph-3*). Sequences were analyzed using MEGA11 (https://www.megasoftware.net/)

(Tamura et al, 2021). All other relevant data and details of resources can be found within the article and its supplementary information.

**First Person**
This article has an associated First Person interview with the first author of the paper.

**Peer review history**
The peer review history is available online at https://journals.biologists.com/bio/lookup/doi/10.1242/bio.062018.reviewer-comments.pdf

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
