## [Peer Review File · Biology Open]

Coincident evolution and functional adaptation of the taxonomically-restricted genes *ivph-3* and *gon-14* in *Caenorhabditis* nematodes

Nikita Jhaveri, Bhagwati Gupta and Helen Chamberlin

DOI: 10.1242/bio.062018

Editor: Sandhya Koushika

Review timeline

Original submission:	16 April 2025
Editorial decision:	9 May 2025
First revision received:	16 September 2025
Accepted:	19 September 2025

Original submission

First decision letter

MS ID#: bio.062018

MS Title: Coincident evolution and functional adaptation of the taxonomically-restricted genes *ivph-3* and *gon-14* in *Caenorhabditis* nematodes.

Authors: Nikita Jhaveri; Bhagwati Gupta; Helen Chamberlin

Article Type: Research Article

Dear Dr Chamberlin,

I have now reached a decision on the above manuscript.

The reviewer reports are shown at the bottom of this email or can be accessed, together with a copy of this decision letter, by going to:

As you will see, the reviewers raised a number of substantial criticisms that prevent me from accepting the paper at this stage.

They suggest, however, that a revised version might prove acceptable, if you can address their concerns. If you think that you can deal satisfactorily with the criticisms on revision, I would be pleased to see a revised manuscript. We would then return it to the reviewers.

Reviewer 1

Comments for the author

In this study, Jhaveri et al., presents an interesting study, where the orthologs between two species have distinct phenotypes and, in some cases, cannot functionally replace each other. This is an example of species-restricted functions of genes that appear to be homologous. The authors presented the critical data in the last Figure, which I think should be the beginning of the study that opens up a stream of experiments to understand the underlying reason for this species-specific

function. I feel that the authors can at least explore whether the functional difference of the two orthologs in the two species are due to expression pattern difference (i.e., promoter sequence evolution) or protein sequence divergence with simple promoter swapping or coding sequencing swapping experiments. For the expression pattern, it is going to be quite feasible, since the single-cell transcriptomic data for both *C. elegans* and *C. briggsae* are available now (see below for the links to these studies). Are there any promoter reporter studies already for these two genes in *C. elegans* and *C. briggsae*?

Comments:

1. There needs to be some description on the potential function of the "RNAse H-fold-containing LIN-15B domain" in the Introduction.
2. Line 95 - Describe what *spr-4* and *htz-1* genes are. For example, what proteins do they code for?
3. Figure 1, some Gene Ontology analysis will be needed to look into the overlapped gene set and the non-overlapped genes.
4. Is *lin-3* also upregulated in *spr-4* and *htz-1* mutants? They can serve as controls.
5. How AlphaSeek was used should be described better. What protein database was searched? What are the hits? (maybe a summary of the hits in a table?) Any structural homologs in species other than humans?
6. Line 148-153, are there synteny conservation for *ivph-3* and *gon-14* among the *Elegans* supergroup species? At minimal, the authors can show that the genomic context of the two paralogs in the gene family is more conserved than other paralogs. This may provide additional evidence for the one-to-one orthology. For example, *lin-15B* may be located in a genomic region (like chromosomal arms or regions with high recombination rate) that is less conserved than the regions containing *ivph-3* and *gon-14*.
7. Figure 3, some representative worm images showing the loss of gonad arms will be needed.
8. Please describe the molecular lesion of the *Cbr-ivph-3* and *Cbr-gon-14* alleles used in this study, so the reader can assess whether they are null alleles or not.
9. Since the phenotypes of *Cbr-ivph-3* and *Cel-ivph-3* are different, it will be good to compare the genes regulated by *Cbr-ivph-3* and *Cel-ivph-3* if the RNA-seq results are available for *ivph-3(-)* mutants in *C. elegans*, although it will require matching the orthologs between *C. elegans* and *C. briggsae*, which I think is not difficult.
10. Figure 4, the experiment is pretty cool. It seems that *Cel-ivph-3* can do more than *Cbr-ivph-3* does and *Cbr-gon-14* carries more functions than *Cel-gon-14*. However, it is not clear whether these additional functions come from protein sequence or regulatory sequences. One possibility is that the protein sequences are interchangeable, but the promoter sequences allow one protein to be expressed in more tissues in one species, thus showing more functions. I think the authors could do experiments to test this hypothesis by using *Cbr-gon-14* promoter to drive *Cel-gon-14* and testing whether that construct can rescue *Cel-gon-14(-)* mutants. Similar things can be done for *ivph-3*. At least, this can narrow down the idea to either promoter divergence or protein sequence divergence. Also, there are single-cell sequencing results for both *C. elegans* and *C. briggsae* now (doi: <https://doi.org/10.1101/2024.11.23.624988>; doi: <https://doi.org/10.1101/2024.02.03.578695>). The authors can at least look into that to see if at the single cell level, whether there are significant differences in expression patterns between the *Cbr* and *Cel* orthologs.
11. Since this gene family is not present in the roots of the *Caenorhabditis* genome, is it possible to look at the genome of species that do not contain any orthologs to figure out whether a *de novo* formation process is involved in making these genes as new genes? For example, by comparing relevant genomic loci of *C. plicata* and *C. bovis*. One may be able to get some ideas about how genes in this family first evolved.
12. The discussion may talk about the potential molecular functions of these genes. Are they involved in RNA stability or chromatin remodeling or something else? How do they regulate other genes?

Minor

Line 62, remove "biological"

Line 71, the use of "stabilizing selection" requires caution, since it is mostly used to refer to one of the three types of trait selection, namely "directional, disruptive, and stabilizing" selection. The authors should not use this term if not referring to specific selection that favors a medium trait value.

Line 227, "may recruit to essential roles"?

Reviewer 2

Comments for the author

This manuscript analyzes the evolution of a small gene family in *Caenorhabditis* nematodes, which encode putative transcriptional repressors including *lin-15B*. The authors previously found in a genetic screen for vulva mutants in *Caenorhabditis briggsae* that mutations in several members of this gene family caused an excess of vulval induction. The present manuscript is a follow up on a previous article "Evolution of Transcriptional Repressors Impacts *Caenorhabditis* Vulval Development" published in 2020.

The authors start by a brief RNAseq analysis of *Cbr-ivph-3* mutants compared with their previous analysis of RNAseq in *Cbr-htz-1* and *Cbr-spr-4* mutants. They then interrogate a blast server for the presence of hits for this genes across the *Caenorhabditis* genus and construct a phylogentic tree of the relationships of the paralogs across species. Using Alphafold, they also propose that the protein structure is analagous to that of human ZBED4. They report on the gonad and sterility phenotype of the mutants in both *C. elegans* and *C. briggsae* and test cross-rescue experiments.

The experiments are sound. I have some comments and suggestions. The main issue is the evolutionary interpretation.

Abstract: The third sentence is obscure and I would remove it- what does "important" mean in "identifying which genes are biologically important"??

Remove the part in the last sentence from "suggest mechanisms". Similarly, the summary statement should not interpret data as showing adaptation.

Discussion: I would remove all sentence from line 227.

Throughout: indicate the temperature at which the experiment was performed.

Figure 1: The *C. briggsae* gene numbers are provided in supplementary tables but some annotation would be welcome.

Figure 2A. The level of divergence of the genes across species is not clear from the figure. It is unclear in the table whether the lack of blast hits indicates divergence or lack of homologs.

Figure 2B. The multiple alignment that forms the basis for this tree would be welcome as a supplement.

Figure 3: Indicate whether all nematode experiments provided in the same graph were performed in parallel.

Note that a defect in gonad arm migration and sterility are phenotypes induced by many mutations and thus not specific. Is the gonad arm defect related to the lack of distal tip cell as in *gon-14* mutants?

Legend: Unclear sentence: " *Cel-gon-14* single and *Cel-ivph-3*; *Cel-gon-14* double mutants exhibit increased variation in the number of induced cells, with doubles statistically different from the wild type. "

idem " is not different from wild type ($p < 0.05$)"

Table S1: typo in first sheet " *Cbr-livp-3*(sy5216)"

Reviewer's Responses to Questions

Experimental quality

Does each figure have the proper controls?

If 'No', please indicate reasons in Comments for Author box below.

Reviewer #1:

- Yes

Reviewer #2:

- Yes

Were the data analyzed using appropriate statistical tests?
If 'No', please indicate reasons in Comments for Author box below.

Reviewer #1:

- Yes

Reviewer #2:

- Yes

Reproducibility

Were experiments performed using adequate number of biological replicates?
If 'No', please indicate reasons in Comments for Author box below.

Reviewer #1:

- Yes

Reviewer #2:

- Yes

Does the methods section provide sufficient detail to permit reproducibility?
If 'No', please indicate reasons in Comments for Author box below.

Reviewer #1:

- Yes

Reviewer #2:

- Yes

Completeness

Are the manuscript's conclusions supported by the data?
If 'No', please indicate reasons in Comments for Author box below.

Reviewer #1:

- No

Reviewer #2:

- No

Scholarship

Do the authors cite and discuss the merits of data that would argue for and against their conclusion?

If 'No', please indicate reasons in Comments for Author box below.

Reviewer #1:

- Yes

Reviewer #2:

- Yes

Does the manuscript title & abstract accurately reflect the contents of the manuscript, without hyperbole?

If 'No', please indicate reasons in Comments for Author box below.

Reviewer #1:

- Yes

Reviewer #2:

- No

First revision

Author response to reviewers' comments

We are grateful to the editor and reviewers for their feedback, and indicate the comments and changes made to address the concerns in italics, below.

Reviewer 1: In this study, Jhaveri et al., presents an interesting study, where the orthologs between two species have distinct phenotypes and, in some cases, cannot functionally replace each other. This is an example of species-restricted functions of genes that appear to be homologous. The authors presented the critical data in the last Figure, which I think should be the beginning of the study that opens up a stream of experiments to understand the underlying reason for this species-specific function. I feel that the authors can at least explore whether the functional difference of the two orthologs in the two species are due to expression pattern difference (i.e., promoter sequence evolution) or protein sequence divergence with simple promoter swapping or coding sequencing swapping experiments. For the expression pattern, it is going to be quite feasible, since the single-cell transcriptomic data for both *C. elegans* and *C. briggsae* are available now (see below for the links to these studies). Are there any promoter reporter studies already for these two genes in *C. elegans* and *C. briggsae*?

Comments:

1. There needs to be some description on the potential function of the "RNase H-fold-containing LIN-15B domain" in the Introduction.

Additional description and references are added to the introduction as suggested.

2. Line 95 - Describe what spr-4 and htz-1 genes are. For example, what proteins do they code for?

Additional description and references are added as suggested.

3. Figure 1, some Gene Ontology analysis will be needed to look into the overlapped gene set and the non-overlapped genes.

This analysis is added as Supplemental Tables 12-15 and discussed in the legend to Figure 1.

4. Is lin-3 also upregulated in spr-4 and htz-1 mutants? They can serve as controls.

lin-3 is indeed upregulated in all three mutants. We have edited the text to more clearly state this observation.

5. How AlphaSeek was used should be described better. What protein database was searched? What are the hits? (maybe a summary of the hits in a table?) Any structural homologs in species other than humans?

The methods have been updated and expanded in the materials and methods, and data are added as Supplemental Tables 16-18.

6. Line 148-153, are there synteny conservation for ivph-3 and gon-14 among the Elegans supergroup species? At minimal, the authors can show that the genomic context of the two paralogs in the gene family is more conserved than other paralogs. This may provide additional evidence for the one-to-one orthology. For example, lin-15B may be located in a genomic region (like chromosomal arms or regions with high recombination rate) that is less conserved than the regions containing ivph-3 and gon-14.

We agree that this is an important question, but addressing it is quite challenging. Part of the reason is that there is variation in the type of data and assembly methods used for the different genomes. This makes assigning a definitive chromosome region and reliably inferring rearrangement breakpoints when synteny is not fully maintained difficult to impossible for many of the genomes. Identifying clear orthologs for evaluating the synteny for family members that do not cluster with ivph-3 or gon-14 is also complicated. For example, while some genome annotation identify a single Ctr-lin-15B gene ([Caendr.org](https://www.ncbi.nlm.nih.gov/nuccore/100881200)), other analyses (Toker et al., 2025) identify that there is not a single 1-to-1 relationship between these species for this gene, which is also apparent in our Figure 2B. This is a big challenge for genes which are not well-conserved across genomes. We have tried to highlight some of the limitations of our approaches for this gene family in the text.

7. Figure 3, some representative worm images showing the loss of gonad arms will be needed.

We have added representative images, as suggested.

8. Please describe the molecular lesion of the Cbr-ivph-3 and Cbr-gon-14 alleles used in this study, so the reader can assess whether they are null alleles or not.

We have added a description of each of the key mutant alleles used in the materials and methods, as suggested.

9. Since the phenotypes of Cbr-ivph-3 and Cel-ivph-3 are different, it will be good to compare the genes regulated by Cbr-ivph-3 and Cel-ivph-3 if the RNA-seq results are available for ivph-3(-) mutants in *C. elegans*, although it will require matching the orthologs between *C. elegans* and *C. briggsae*, which I think is not difficult.

We agree that this would be a good experiment to do. Unfortunately, there are not RNA-seq data available for Cel-ivph-3(-), so we were not able to complete this analysis.

10. Figure 4, the experiment is pretty cool. It seems that Cel-ivph-3 can do more than Cbr-ivph-3 does and Cbr-gon-14 carries more functions than Cel-gon-14. However, it is not clear whether these additional functions come from protein sequence or regulatory sequences. One possibility is that the protein sequences are interchangeable, but the promoter sequences allow one protein to be expressed in more tissues in one species, thus showing more functions. I think the authors could do experiments to test this hypothesis by using Cbr-gon-14 promoter to drive Cel-gon-14 and testing whether that construct can rescue Cel-gon-14(-) mutants. Similar things can be done for ivph-3. At least, this can narrow down the idea to either promoter divergence or protein sequence divergence. Also, there are single-cell sequencing results for both *C. elegans* and *C. briggsae* now (doi: <https://doi.org/10.1101/2024.11.23.624988>; doi: <https://doi.org/10.1101/2024.02.03.578695>). The authors can at least look into that to see if at the single cell level, whether there are significant differences in expression patterns between the Cbr and Cel orthologs.

We appreciate the reviewer's comments, and also find the gene differences interesting. To address the comment, we have reviewed the data from Toker et al., 2025, and Large et al., 2025, excerpted their data in Supplemental Table 19, and discussed the observations in the text. We have also produced chimeric transgenes that include the sequences upstream of the start codon from one species and the sequences downstream from the other, and evaluated the capacity of these transgenes to rescue the mutants. These data are added to Figure 4 (panel C and E). In the text, we specifically consider the hypothesis that the gene rescue experiments result from difference in gene expression - for example that the genes that can rescue in only one species exhibit more restricted expression than the ones that rescue in both. Notably, the data do not support this hypothesis, and the chimeric gene experiments suggest that the functional differences may either be more distributed across the gene (for Cbr-gon-14) or not limited to upstream sequences (for Cel-ivph-3). We acknowledge that a more direct answer to the question will require further experiments, and we plan to address this in future work.

11. Since this gene family is not present in the roots of the Caenorhabditis genome, is it possible to look at the genome of species that do not contain any orthologs to figure out whether a de novo formation process is involved in making these genes as new genes? For example, by comparing relevant genomic loci of *C. plicata* and *C. bovis*. One may be able to get some ideas about how genes in this family first evolved.

*We agree that this is an important question, but addressing it is quite challenging. Part of the reason is that there is variation in the type of data and assembly methods used for the different genomes. Another piece is the divergence (or sparse representation) of the species/genomes at the roots of the genus. A brief check using the current assemblies suggests the genes flanking CBOVI.g10510 and CBOVI.g3374 are not adjacent to each other in *C. plicata* or *C. parvicauda* genomes, but the quality of the genome assemblies and the divergence make it difficult to identify breakpoints or infer evolutionary trajectories. We see this as an important question to address in future work.*

12. The discussion may talk about the potential molecular functions of these genes. Are they involved in RNA stability or chromatin remodeling or something else? How do they regulate other genes?

While the molecular functions for IVPH-3 and GON-14 are not yet understood, we have added some additional points to the discussion to address the comment.

Minor

Line 62, remove "biological"

Removed as suggested.

Line 71, the use of "stabilizing selection" requires caution, since it is mostly used to refer to one of the three types of trait selection, namely "directional, disruptive, and stabilizing" selection. The authors should not use this term if not referring to specific selection that favors a medium trait value.

This sentence has been edited as suggested.

Line 227, "may recruit to essential roles"?

This sentence has been removed at the suggestion of reviewer 2.

Reviewer 2: This manuscript analyzes the evolution of a small gene family in *Caenorhabditis* nematodes, which encode putative transcriptional repressors including *lin-15B*. The authors previously found in a genetic screen for vulva mutants in *Caenorhabditis briggsae* that mutations in several members of this gene family caused an excess of vulval induction. The present manuscript is a follow up on a previous article "Evolution of Transcriptional Repressors Impacts *Caenorhabditis* Vulval Development" published in 2020.

The authors start by a brief RNAseq analysis of *Cbr-ivph-3* mutants compared with their previous analysis of RNAseq in *Cbr-htz-1* and *Cbr-spr-4* mutants. They then interrogate a blast server for the presence of hits for this genes across the *Caenorhabditis* genus and construct a phylogentic tree of the relationships of the paralogs across species. Using AlphaFold, they also propose that the protein structure is analogous to that of human ZBED4. They report on the gonad and sterility phenotype of the mutants in both *C. elegans* and *C. briggsae* and test cross-rescue experiments.

The experiments are sound. I have some comments and suggestions. The main issue is the evolutionary interpretation.

Abstract: The third sentence is obscure and I would remove it- what does "important" mean in "identifying which genes are biologically important"??

This sentence has been removed as suggested.

Remove the part in the last sentence from "suggest mechanisms". Similarly, the summary statement should not interpret data as showing adaptation.

These have been removed from the abstract and summary statement, as suggested.

Discussion: I would remove all sentence from line 227.

The speculative interpretation has been removed from the discussion, as suggested.

Throughout: indicate the temperature at which the experiment was performed.

We have added a comment about growth conditions to the materials and methods section, as suggested.

Figure 1: The *C. briggsae* gene numbers are provided in supplementary tables but some annotation would be welcome.

As also requested by reviewer 1, we have added further discussion including GO term enrichment to the figure legend and GO term enrichment data as Supplemental Tables 12-15 to address the concern. We have also added more gene information (Cbr- gene name and WB gene ID) to Supplemental Tables 3-5 to improve a reader's ability to review the lists for genes of interest.

Figure 2A. The level of divergence of the genes across species is not clear from the figure. It is unclear in the table whether the lack of blast hits indicates divergence or lack of homologs.

We have altered 2A to better illustrate the relationships among the species, and edited the text to clearly indicate that lack of blastp hit does not mean lack of homologs. We have also more clearly described the Foldseek analysis (and its limitations) and results in materials and methods (Supplemental Tables 16-18). We agree that it is very challenging to definitively conclude whether lack of blast hits indicates divergence or lack of homologs.

Figure 2B. The multiple alignment that forms the basis for this tree would be welcome as a supplement.

The 77 sequences used in the analysis are included as Supplemental Table 9, and we have added the MEGA .msx session file that includes the alignment as an additional Supplemental File to address the comment.

Figure 3: Indicate whether all nematode experiments provided in the same graph were performed in parallel.

All experiments in a given graph were conducted in parallel. A comment confirming this has been added to each experiment in the materials and methods section.

Note that a defect in gonad arm migration and sterility are phenotypes induced by many mutations and thus not specific. Is the gonad arm defect related to the lack of distal tip cell as in gon-14 mutants?

The mutants appear to lack a distal tip cell, although molecular markers were not used. To address the comment, we have added representative images to Figure 3, as requested by Reviewer 1.

Legend: Unclear sentence: " Cel-gon-14 single and Cel-ivph-3; Cel-gon-14 double mutants exhibit increased variation in the number of induced cells, with doubles statistically different from the wild type. "idem " is not different from wild type ($p < 0.05$)"

We have edited the figure legend to more clearly state the observation.

Table S1: typo in first sheet " Cbr-livp-3(sy5216)"

The table has been corrected, as suggested.

Second decision letter

MS ID#: bio.062018R1

MS Title: Coincident evolution and functional adaptation of the taxonomically-restricted genes ivph-3 and gon-14 in Caenorhabditis nematodes.

Authors: Nikita Jhaveri; Bhagwati Gupta; Helen Chamberlin

Article Type: Research Article

Dear Dr Chamberlin,

I am happy to tell you that your manuscript has been accepted for publication in Biology Open, pending our standard publication integrity checks. It was accepted on 19 September 2025.